

# Classification of *Zophobas morio* and *Tenebrio molitor* using transfer learning

Agus Pratondo[1] and Arif Bramantoro[2]

[1] School of Applied Sciences, Telkom University, Bandung, West Java, Indonesia
[2] School of Computing and Informatics, Universiti Teknologi Brunei, Bandar Seri Begawan, Brunei Darussalam

## ABSTRACT

*Zophobas Morio* and *Tenebrio Molitor* are popular larvae as feed ingredients that are widely used by animal lovers to feed reptiles, songbirds, and other poultry. These two larvae share a similar appearance, however; the nutritional ingredients are significantly different. *Zophobas Morio* is more nutritious and has a higher economic value compared to *Tenebrio Molitor*. Due to limited knowledge, many animal lovers find it difficult to distinguish between the two. This study aims to build a machine learning model that is able to distinguish between the two. The model is trained using images that are taken from a standard camera on a mobile phone. The training is carried on using a deep learning algorithm, by adopting an architecture through transfer learning, namely VGG-19 and Inception v3. The experimental results on the datasets show that the accuracy rates of the model are 94.219% and 96.875%, respectively. The results are quite promising for practical use and can be improved for future works.

## INTRODUCTION

Computer vision has been used in various fields of life, such as agriculture, animal husbandry, health, smart cities, and others (*Pratondo et al., 2014*; *Rizqyawan et al., 2020*). In agriculture and animal husbandry, the use of computer vision for the classification and detection of various objects has been widely practiced (*Abd Aziz et al., 2021*; *Thai, Nguyen & Pham, 2021*). The detection and classification of similar objects are challenging tasks for researchers to provide the best possible accuracy.

*Zophobas Morio* and *Tenebrio Molitor* are two kinds of larva that share the same morphology. However, *Zophobas Morio* is more nutritious compared to *Tenebrio Molitor* (*Purnamasari et al., 2018*; *Santoso, Afrila & Fitasari, 2017*). These nutritional advantages from *Zophobas Morio* make it preferable for animal lovers. *Zophobas Morio* can be used as feed for various animals, especially for chirping birds. The comparison of the two larvae in term of their ingredients is presented in Table 1 as previously studied in *Benzertiha et al. (2019)*. It can be seen that *Zophobas Morio* has more nutrition ingredients, such as dry matter, protein, and ether extract; than *Tenebrio Molitor* has. Only chitin is the exception.

Because of the similar morphology of the two larvae, people with limited knowledge are often unable to distinguish between them. Several buyers may spend money as high

Corresponding author
Agus Pratondo,
pratondo@telkomuniversity.ac.id

| Table 1 | Nutrition ingredients for the two larvae (*Benzertiha et al., 2019*). | |
| --- | --- | --- |
| Item | *Tenebrio Molitor* | *Zophobus Morio* |
| Dry matter (DM, %) | 95.58 | 96.32 |
| Crude protein (% of DM) | 47.0 | 49.3 |
| Ether extract (% of DM) | 29.6 | 33.6 |
| Chitin (% of DM) | 89.1 | 45.9 |

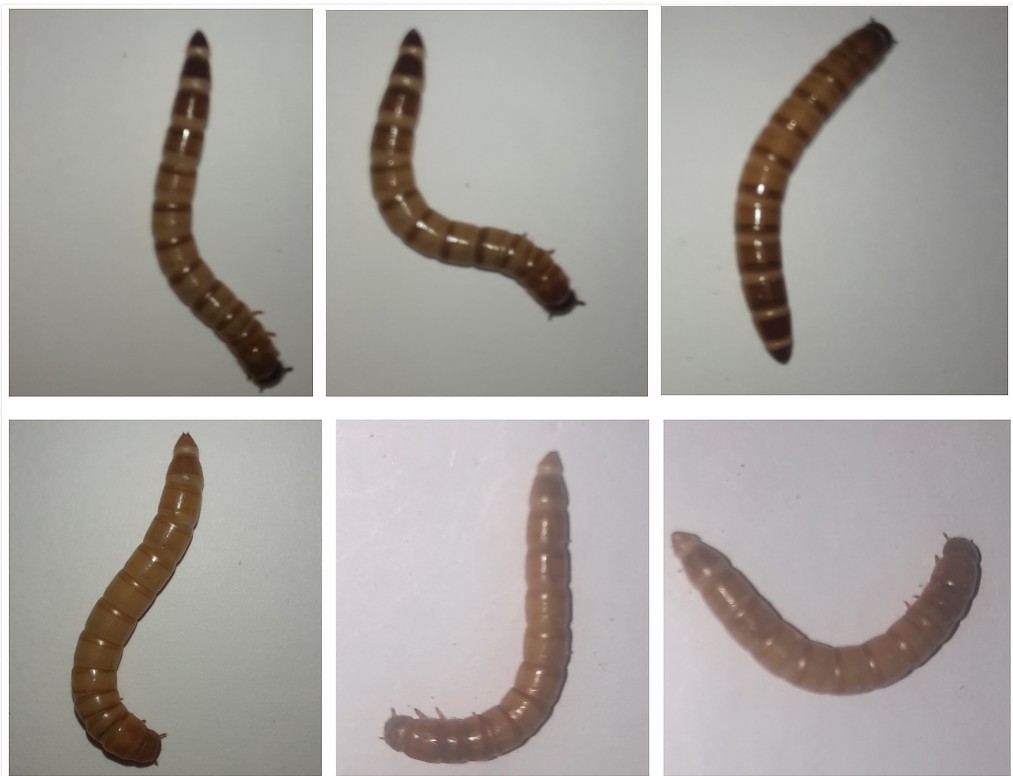

**Figure 1** Images of *Zophobas Morio* (upper) and *Tenebrio Molitor* (lower).

as the price of *Zophobas Morio*; however, the purchased larva is actually *Tenebrio Molitor*. Figure 1 shows several samples of *Zophobas Morio* and *Tenebrio Molitor* images.

This study aims to build an application that is able to distinguish *Zophobas Morio* and *Tenebrio Molitor* larvae based on images obtained from mobile phone cameras. The images are then analyzed for further classification based on the model which is built using a machine learning algorithm. Through the model, the difficulty of distinguishing the two larvae can be resolved.

The remainder of this paper will discuss how the model is developed and evaluated. In section 2, a literature review will be discussed for the methods that are related and used to build the classification model. In section 3, we will discuss the experiments carried out which include dataset preparation, experimental settings, and experimental results. In section 4, discussions on the experimental results and the possibility of further research in

the future will be elaborated. Section 5 will provide more related works in detail. Finally, conclusions will be presented in section 6.

## METHODS

Traditionally, image classification can be performed by manually selecting features. Features are designed and extracted from training images. However, selecting features is a complicated task and often requires a high expertise. To date, the use of pixels as features is widely employed for image classification. Original classification algorithms, such as the Naive Bayesian, $k$-nearest neighbors($k$-NN), and support vector machines (SVM), are commonly used before the era of deep learning, which is basically developed from the artificial neural networks (*Bishop, 2006*). These traditional classification algorithms, nevertheless, are still implemented in baseline experiments with classification tasks.

This study employs transfer learning which is an advanced development of the artificial neural networks method. The theoretical background is presented successively from artificial neural networks, followed by deep learning, and finally, transfer learning.

### Artificial neural networks

The artificial neural network is a mathematical model of problem-solving inspired by the functioning of human nerves (*Bishop, 1995*; *Haykin, 2010*). The simplest model of an artificial neural network is a single-layer perceptron, which is diagrammatically presented in Fig. 2.

Mathematically, the single-layer perceptron is expressed as follows:

$$\sum_{x=1}^{x=n} w_i \mathbf{x}_i + b \tag{1}$$

where $w_i$, $x_i$ and $b$ denote weight $-i$, feature $-i$ and bias consecutively. The single perceptron receives information on attribute/feature values and assigns a weight to each feature. The final result is determined by passing the sum of the values to the activation function. Several commonly used activation functions are sigmoid, hyperbolic tangent, and rectified Linear Unit (ReLU).

### Deep neural networks

The artificial neural network continues to develop. Researchers proceed to model it as a multilayer perceptron (*Goodfellow, Bengio & Courville, 2016*). Similar to the single-layer perceptron, the multilayer perceptron receives input in the form of a set of feature values to then be weighted and the final result is entered into the activation function. The difference between single-layer perceptron and multilayer perceptron is that there is a hidden layer that automatically adds the number of nodes and creates more complex connections. Hence, multilayer perceptron can solve any problems that cannot be solved by using single-layer perceptron.

The use of multilayer perceptron continues to grow, especially in the field of computer vision. Hardware support enables more complex image operations, such as convolution procedures. With the use of artificial neural networks that have many hidden layers and

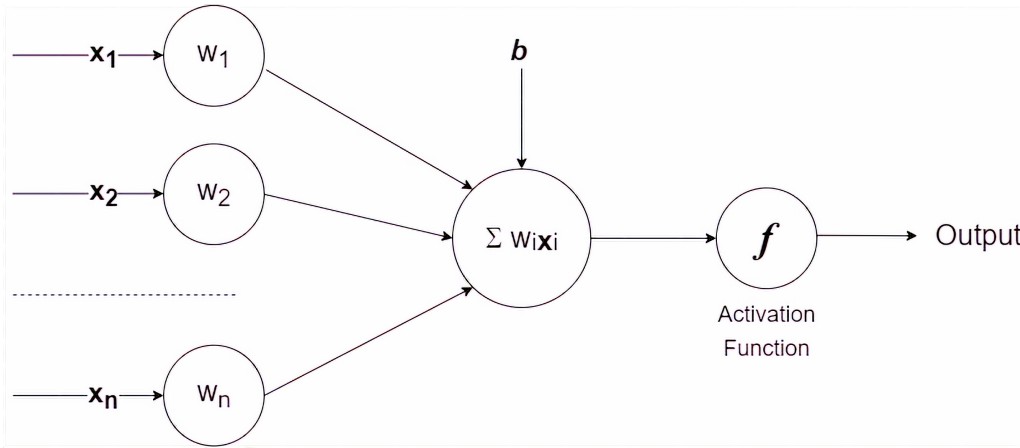

**Figure 2** A single layer perceptron.

complex image operations, the era of deep neural networks that is more popularly known as deep learning begins.

## Related works on deep learning

The use of deep learning in classifying a collection of images becomes more popular due to the explosion of data and the availability of the high-end computational platforms. The classification of larvae images takes the advantage of the deep learning platform. To the best of our knowledge, however, only a few studies have been proposed in this field.

The first related work is proposed in *Fuad et al. (2018)* that utilized transfer learning technique, *i.e.,* Google inception model (*Szegedy et al., 2017*), to identify the larvae of *Aedes Aegypti* which are commonly found inside the clean liquid place. These larvae are important to classify because it helps reduce the spread of dengue fever, especially in tropical countries. There were enough pictures, *i.e.,* more than five hundred, used in this paper due to the assorted canvas and lightning within the liquid place. The paper mentioned that the dataset was produced by laboratory work, however, it remains unclear why the authors chose to have no data pre-processing required by the study. It is common to have data pre-processing as an essential step in deep learning especially dealing with real-world pictures, although most researchers skip the explanation in their reports. The authors claimed that the less the learning rate is the better the classification accuracy. It can be understood because there are only two classes are produced, *i.e.,* larva or not larva. In detail, the accuracy in the paper achieved 99.98% and 99.90% for learning rate 0.1, 99.91% and 99.77% for learning rate 0.01; and 99.10% and 99.93% for learning rate 0.001. The cross-entropy error in the paper achieved 0.0021 and 0.0184 for learning rate 0.1, 0.0091 and 0.0121 for learning rate 0.01; and 0.0513 and 0.0330 for learning rate 0.001. Although the metrics of accuracy and cross-entropy errors are presented, the significance of the experiments could not be justified due to the goal of the paper is to provide informative accuracies and learning errors with three learning rates. It would be interesting to see the baseline of the accuracy performed by this paper with different kinds of the larva.

*Aedes Aegypti* is the most popular larva to be deeply classified from its images. Another work on this larva is proposed in *Azman & Sarlan (2020)* that examined whether particular water storage is the suitable place for the mosquitoes to lay their larva or not. Although it is not explicitly mentioned, it can be assumed that the paper provides two different types of larva from the mosquitoes that spread the lethal epidemic of dengue fever and three other types of larvae from different mosquitoes. These five types of the larva are sorted based on the prediction accuracy by utilizing a general convolution neural network algorithm, although there is no further explanation in detail on how to implement this algorithm with specific techniques. The result shows the unbalanced accuracies for each type of larvae that varies from 0.7% to 73%. It remains unclear why the gap between the two types of dengue fever mosquito larva and three other types is huge.

The authors in *Asmai et al. (2019)* also analyzed *Aedes Aegypti* larvae. The testing was conducted from ten images of *Aedes Aegypti* larvae and ten images of non *Aedes Aegypti* larvae. Various methods of convolution neural networks were utilized, such as VGG16 (*Zhang et al., 2015*), VGG-19 (*Zhang et al., 2015*), ResNet-50 (*Koonce, 2021*), and InceptionV3 (*Szegedy et al., 2017*). The results showed that Resnet-50 outperforms other methods in terms of the implementation on mobile devices, although its performance in terms of accuracy and loss was lower than VGG-19 (*Zhang et al., 2015*). However, combining the mobile performance from two regular metrics, such as accuracy and loss, with other provided metrics, such as file size and training time, should be further investigated because it is not commonly used in deep learning.

A different kind of larva is analyzed in *Shang, Lin & Cong (2020)* that addressed the issue of a low number of pictures with high-quality tags. Another issue tackled by this paper was that the picture of *Zebrafish* larva usually contains the fuzzy classification for ten types of the larva, such as deceased, regular, and short bottom. Hence, the deep learning is non-trivial in classifying this larva. The result showed that there was an improvement of classification performance between the proposed two-layered classification technique and other deep learning methods, such as GoogLeNet (*Szegedy et al., 2015*), VGG-19 (*Zhang et al., 2015*) and AlexNet (*Krizhevsky, Sutskever & Hinton, 2012*), by reproducing the previous researches with the same dataset. The improvement compared to the baseline reached 22%. The obtained accuracy mean was 91% for overall types and the obtained maximum accuracy was 100% for deceased and skin types. However, it remains unclear how to obtain these numbers since there is no exact numbers provided by the paper. Moreover, it should have been more interesting to see how the technique operates, since running two layers of classification in a single workflow is obviously better than running one layer, regardless of the image characteristics.

The latest work from aquaculture researchers in *Kakehi et al. (2021)* presented the use of deep learning in identifying oyster larvae. The work was important for the oyster farmers to speed up the process of larva identification during the oyster growing time. The result was claimed to have high performance with more than 80% for precision, around 90% or recall, and therefore around 86% for F-score. However, the overall process was not completely automatic, since human intervention was required during the process of identification. It is interesting to note that the difficulty was due to the characteristics of oyster larva

that occasionally mixed with other objects that produce several layers on the images. This paper also provided three-dimension graphs of shell height, long side, and short side to help understand the dataset. These three measures were estimated by using the coordinate system of PyTorch (*Ketkar, 2017b*).

Another latest work was proposed in *Ong, Ahmad & Majid (2021)* that provided a deep learning model to predict the teemingness of the larvae of house flies. The experiments were conducted with different angles of picture shooting. The variation also includes various colors and textures. The final result of the proposed method performance was precision between 88.44% and 92.95%, recall between 88.23% and 94.10%, accuracy between 87.56% and 92.89%, and F-score between 88.08% and 93.02%. Another interesting result from this paper is that the image with green and white lighting performed the best while the images with red lighting performed the worst. However, it should be further researched whether the result from this larva remains the same as the results from other kinds of larvae. Another critical thinking on this paper is that the benefits of the classification result on house fly larva to the real world remain questionable, except for the sake of experimental laboratory. In addition, it would be more interesting to see the specific technique of deep learning used in the paper instead of explaining convolution neural networks in general.

Overall, the presented literature review shows that the use of deep learning in classifying larva images is eminent. However, none of the papers, to the best of our knowledge, handled the images of *Zophobas Morio* and *Tenebrio Molitor* larvae. These larvae are important for feeding broiler chicken in emerging countries; hence, the classification of the larvae can be an additional feature for other presented larva automatic detection systems. This is also due to the fact that the detection for two different larvae by naked eyes has lower accuracy as been identified by biologists (*Benzertiha et al., 2019*).

## Transfer learning

Several transfer learning algorithms have been proposed, including VGG-19, Resnet-50, and InceptionResnetV2 (*Goodfellow, Bengio & Courville, 2016*; *Simonyan & Zisserman, 2014*; *He et al., 2016*), to enhance deep learning with less amount but significant enough dataset. Instead of using deep learning from scratch for the two specific larvae, transfer learning is more suitable because the previous works have been proposed for various larvae.

VGG-19 is a more advanced development of VGG-16 (*Zhang et al., 2015*). It has 19 layers, which is quite a contrast compared to Resnet-50 that has 50 layers, and InceptionResnetV2 which that 164 layers. The architecture for VGG-19 can be seen in Fig. 3.

For practical purposes, two pre-trained models are selected in the experiment, namely, VGG-19 and Inception v3. The first model has lower layers than the second one has.

Since there is no previous research related to the classification of *Zophobas Morio* and *Tenebrio Molitor* larvae classification, two traditional classification algorithms, specifically, *k*-NN and SVM, are employed (*Géron, 2019*). The results from these two algorithms are used as a reference for the baseline. In other words, the result of the pre-trained models is compared to the traditional classifiers.

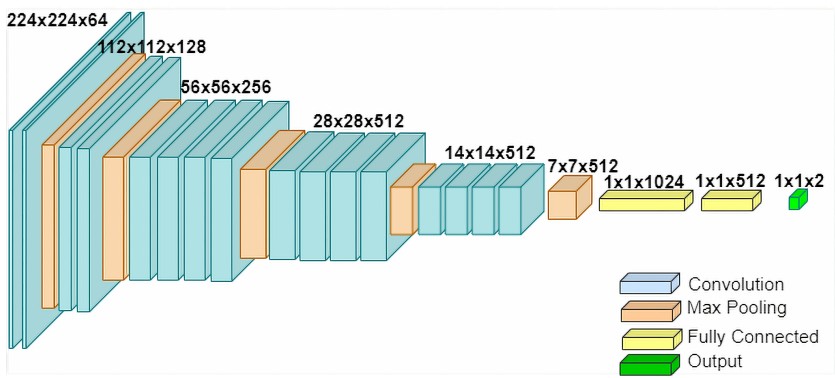

**Figure 3** The architecture of VGG-19.

## RESULTS

Research preparation, especially how to get the dataset, and parameter settings on the model, as well as the experimental results of the model on the dataset are presented as follows.

### Datasets

The datasets are obtained using a standard mobile phone camera commonly available on the market. In this experiment, we use a mobile phone with specifications: Memory 6 GB, Processor Exynos 9611 Octa-core, Camera: Quad camera 48 MP+12 MP ultrawide+5 MP macro+5 MP depth. The images were taken with the use of flash. These specifications are important for reproducing the research in the future.

Larva image samples were obtained from the animal market. The captured larvae are expected to be alive, although several larvae are generally mixed with skins and dead larvae. The total number of samples was 640 equally distributed for each *Zophobas Morio* and *Tenebrio Molitor*. Each sample was placed on a white paper sheet and subsequently captured with a distance of approximately 10 centimeters from the larvae as previously displayed in Fig. 1. Since the images are designed with homogeneous background, *i.e.,* white paper, and the size of the larva is various, it is not necessary to perform an image augmentation on the dataset.

The dataset is divided into 10 folds during the experiment. Each fold is chosen as a testing set while the remaining folds are utilized as a training set. Moreover, the distribution of each class in the fold is considered identical for all folds. This split is known as stratified cross-validation (CV) (*Breiman et al., 2017*). We summarize the distribution of the dataset in Table 2. Each class consists of 320 images which is divided into 10 folds. One fold containing 32 images per class is assigned as data testing while the remaining ones are as data training. It is also important to note that each fold is assigned as testing data only once. After 10 folds are iteratively evaluated, the testing set will completely contain 320 images.

**Table 2  Image distribution for 10-CV.**

| Species | Total | Training per CV | Testing per CV |
|---|---|---|---|
| *Zophobas Morio* | 320 | 288 | 32 |
| *Tenebrio Molitor* | 320 | 288 | 32 |

## Model implementation

We carried out experiments for the above datasets using deep learning. The model used is taken from another model that has been tested and has the same objective characteristics, namely image classification. As stated in section 2, two transfer learning methods are employed in building the classification model, namely VGG-19 (*Zhang et al., 2015*) and Inception v3 (*Szegedy et al., 2016*).

The transfer learning using VGG-19 is implemented in Python programming language and run on Google Colab (*Paper, 2021*). The library for implementing the transfer learning is Keras (*Gulli & Pal, 2017*) which is imported from Tensorflow (*Abadi et al., 2016*). In VGG-19, the number of layers has been defined as 19. The number of layers is fixed for this model. However, there are still several parameters that we set based on the number of datasets used. First, the trainable parameter is set to false because the network will not be trained again. Then, the last classification layer is set to 2 because in this experiment there are only two labels, namely *Zophobas Morio* and *Tenebrio Molitor*. On the initial layer, the image is adjusted to the size of $(224 \times 224 \times 3)$. The batch size is set to 32 while the epoch is 25.

Similarly, Inception v3 is implemented in Python and executed on Google Colab. Inception v3 has 48 layers and the trainable parameter is set to false. In order to provide a fair comparison, the batch and the epoch parameters are set to the same setting as VGG-19.

## Experimental results

The testing data are utilized in the final evaluation to determine the performance of the built model. As previously mentioned, two traditional classification algorithms, namely $k$-NN and SVM, are selected. The results serve as a baseline because no previous study exists on the classification of the two larvae to the best of our knowledge.

The detailed results for $k$-NN with various $k$ values are listed in Table 3. The image size is scaled down to $224 \times 224$. The value of $k$ is set at an odd number since the classification is binary. The results of $k$-NN and SVM are used as the baseline. The best results on a particular $k$ in $k$-NN are chosen and subsequently considered a reliable representation of $k$-NN in general.

Similar to $k$-NN, SVM is performed on resized images, *i.e.,* $224 \times 224$. The selected kernel is linear due to its simplicity. The results of SVM, as well as $k$-NN, are subsequently compared to the two transfer learning algorithms used in this research, *i.e.,* VGG-19 and Inception v3. The comparison of accuracy per fold between $k$-NN, SVM, VGG-19, and Inception v3 is presented in Table 4. Please note that the assignment of $k = 1$ for the $k$-NN is obtained from the previous experimental result as listed in Table 3

**Table 3** Experimental results using k-NN.

| Testing fold | Training fold | Accuracy | | | | | Average |
|---|---|---|---|---|---|---|---|
| | | k=1 | k=3 | k=5 | k=7 | k=9 | |
| 1 | 2-9 | 90.625 | 84.375 | 82.812 | 84.376 | 76.562 | **83.750** |
| 2 | 1,3-10 | 84.375 | 81.250 | 79.688 | 76.562 | 71.875 | 78.750 |
| 3 | 1-2,4-10 | 81.250 | 81.250 | 76.562 | 76.562 | 75.000 | 78.125 |
| 4 | 1-3,5-10 | 81.250 | 82.812 | 82.812 | 76.562 | 75.000 | 79.687 |
| 5 | 1-4,6-10 | 82.812 | 76.562 | 71.875 | 73.438 | 68.750 | 74.687 |
| 6 | 1-5,7-10 | 82.812 | 78.125 | 75.000 | 75.000 | 71.875 | 76.562 |
| 7 | 1-6,8-10 | 75.000 | 73.438 | 68.750 | 67.188 | 65.625 | 70.000 |
| 8 | 1-7,9-10 | 92.188 | 81.250 | 75.000 | 73.438 | 71.875 | 78.750 |
| 9 | 1-8,10 | 89.062 | 82.812 | 79.688 | 79.688 | 76.562 | 81.562 |
| 10 | 1-9 | 75.000 | 78.125 | 76.562 | 75.000 | 70.312 | 75.000 |
| | Average | **83.437** | 80.000 | 76.875 | 75.781 | 72.344 | 77.687 |

**Table 4** Accuracy of various models.

| Testing fold | Training fold | Accuracy | | | |
|---|---|---|---|---|---|
| | | k-NN | SVM | VGG-19 | Inception v3 |
| 1 | 2-9 | 90.625 | 96.875 | 89.625 | 98.438 |
| 2 | 1,3-10 | 84.375 | 95.313 | 95.313 | 98.438 |
| 3 | 1-2,4-10 | 81.250 | 92.188 | 98.438 | 96.875 |
| 4 | 1-3,5-10 | 81.250 | 92.188 | 95.313 | 100.000 |
| 5 | 1-4,6-10 | 82.812 | 90.625 | 96.875 | 92.188 |
| 6 | 1-5,7-10 | 82.812 | 95.313 | 95.313 | 98.438 |
| 7 | 1-6,8-10 | 75.000 | 93.750 | 92.188 | 89.438 |
| 8 | 1-7,9-10 | 92.188 | 95.313 | 90.625 | 89.063 |
| 9 | 1-8,10 | 89.062 | 87.500 | 90.625 | 98.438 |
| 10 | 1-9 | 75.000 | 90.625 | 95.313 | 98.438 |

## DISCUSSIONS

The main objective of the proposed classification model is to determine whether an observed larva image is categorized as a *Zophobas Morio* or not. Since there are only two classes, the model can be considered a binary classification. Hence, *Zophobas Morio* can be viewed as a positive class while *Tenebrio Molitor* can be a negative class.

There are several available performance metrics in binary classification. However, we prefer to use the most common ones, such as precision, recall, and accuracy. Let $TP, TN, FP$, and $FN$ be True Positive, True Negative, False Positive, and False Negative, respectively. Precision, recall, and accuracy can be expressed as:

$$Precision = \frac{TP}{TP + FP}, \tag{2}$$

$$Recall = \frac{TP}{TP + FN}, \tag{3}$$

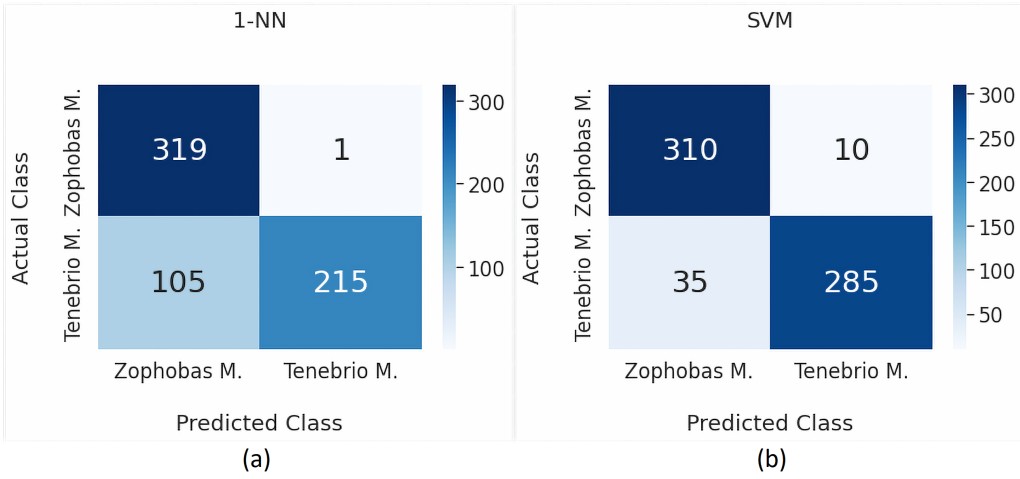

**Figure 4** Classification results using *k*-NN and SVM.

$$Accuracy = \frac{TP + TN}{TP + FP + FP + FN} \times 100\%. \tag{4}$$

By employing Eqs. (2), (3) and (4), the performance for each model are visually expressed in a confusion matrix as shown in Figs. 4A and 4B, 5A and 5B consecutively. Table 5 provides a more detailed performance comparison by including precision, recall, and accuracy. It can be seen that the chosen transfer learning models outperform the commonly used traditional models. Moreover, the best performance is achieved by the Inception v3 model where the precision and recall are 0.972 and 0.966 respectively, while the accuracy is 96.875%.

The highest precision value is 0.972, which means that 97.2% of positive predictions are true. Meanwhile, the recall value is 0.966, which indicates that 96.6% of positive data are correctly detected. We are quite confident of the positive prediction result obtained in this research. Furthermore, the final result of accuracy is 96.875%. This high accuracy score is significantly promising for future practical use. Especially, when these results are compared to other studies previously reviewed. Table 6 compares the previously reviewed papers to our approach. It can be argued that our work outperforms the others in term of general performance metrics.

Several techniques can be applied to improve the existing performance in the future. An alternative of general solution for the classification problem is usually to increase the dataset. It is expected that the addition of the dataset can increase the number of recognizable patterns. Another alternative is to use more complex transfer learning methods, such as Resnet-50 (*Koonce, 2021*) and Efficient Nets model (*Tan & Le, 2020*). Compared to those models, the methods of VGG-19 and Inception v3 used in this study have a simpler architecture, and therefore, they can be used as a baseline of transfer learning approach for the two larvae research.

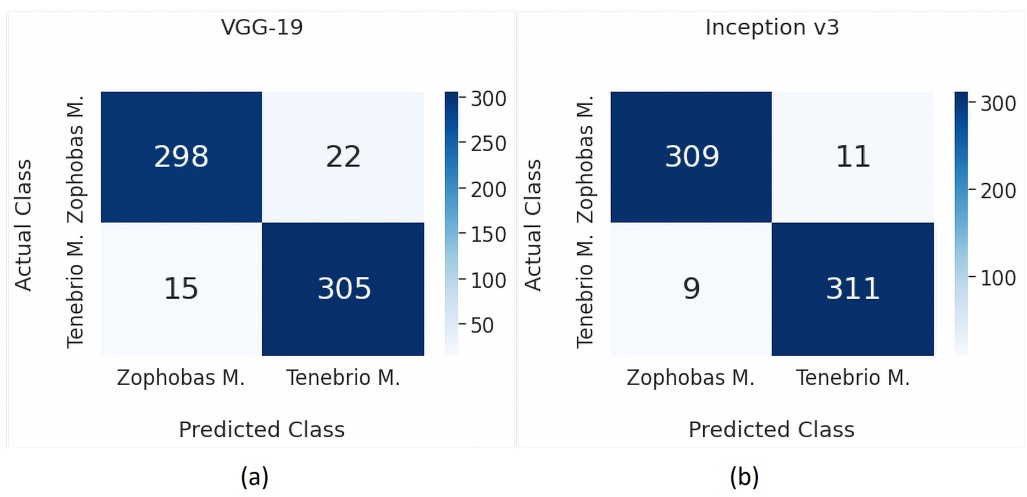

**Figure 5** Classification results using VGG-19 and Inception v3.

**Table 5  Performance metrics for binary classification.**

| Method | TP | FP | TN | FN | Precision | Recall | Accuracy |
|---|---|---|---|---|---|---|---|
| k-NN | 319 | 105 | 215 | 1 | 0.752 | 0.997 | 83.438 |
| SVM | 310 | 35 | 285 | 10 | 0.899 | 0.969 | 92.969 |
| VGG-19 | 298 | 15 | 305 | 22 | 0.952 | 0.931 | 94.219 |
| Inception v3 | 309 | 9 | 311 | 11 | 0.972 | 0.966 | 96.875 |

**Table 6  Related works comparison.**

| Work | Larva types | Methods | Results |
|---|---|---|---|
| (*Fuad et al., 2018*) | *Aedes Aegypti* | Google inception model | 99.77–99.98% accuracy, 0.21–5.13% cross-entropy error |
| (*Azman & Sarlan, 2020*) | *Aegypti, Albopictus, Anopheles, Armigeres, Culex* | Convolution neural network | 0.7–73% accuracy |
| (*Asmai et al., 2019*) | *Aedes Aegypti* | VGG16, VGG-19, ResNet-50, InceptionV3 | 77.31–85.10% accuracy, 0.31–0.66% loss |
| (*Shang, Lin & Cong, 2020*) | *Zebrafish* | GoogLeNet, VGG-19, AlexNet | 91–100% accuracy |
| (*Kakehi et al., 2021*) | Oyster | coordinate system of PyTorch | 82.4% precision, 90.8% recall, 86.4% F-score |
| (*Ong, Ahmad & Majid, 2021*) | House flies | Convolution neural network | 88.44–92.95% precision, 88.23–94.10% recall, 87.56–92.89% accuracy, 88.08–93.02% F-score |
| Ours | *Zophobas Morio, Tenebrio Molitor* | VGG-19, Inception v3 | 97.2% precision, 96.6% recall, 96.876% accuracy |

## CONCLUSIONS

We built the complete models for the classification of *Zophobas Morio* and *Tenebrio Molitor* larvae using two transfer learning methods, namely VGG-19 and Inception v3. These models outperform the traditional ones, *i.e., k*-NN and SVM. The experimental

results show that Inception v3 achieves the best accuracy, *i.e.,* 96.875%. In addition, the values of precision and recall are 0.972 and 0.966, respectively. These results are quite promising for practical use. Several improvements to this model can still be made for further research, such as increasing the dataset and combining more advanced transfer learning methods.

### Funding

This work is supported by Telkom University Publication Grant and Universiti Teknologi Brunei sponsorship. The funders had no role in study design, data collection and analysis, decision to publish, or preparation of the manuscript.

### Grant Disclosures

The following grant information was disclosed by the authors:
Telkom University Publication Grant.
Universiti Teknologi Brunei sponsorship.

### Competing Interests

The authors declare there are no competing interests.

### Author Contributions

- Agus Pratondo conceived and designed the experiments, performed the experiments, analyzed the data, performed the computation work, prepared figures and/or tables, authored or reviewed drafts of the paper, and approved the final draft.
- Arif Bramantoro performed the experiments, analyzed the data, prepared figures and/or tables, authored or reviewed drafts of the paper, and approved the final draft.

### Data Availability

   The data is available at figshare: Pratondo, Agus; Bramantoro, Arif (2021): larva-20211102T160758Z-001.zip. figshare. Dataset. https://doi.org/10.6084/m9.figshare.16918873.v1.

### Supplemental Information

Supplemental information for this article can be found online at http://dx.doi.org/10.7717/peerj-cs.884#supplemental-information.

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
