# Peer review of "Classification of Zophobas morio and Tenebrio molitor using transfer learning"

_PeerJ Computer Science, doi:10.7717/peerj-cs.884_

## Round 0.1 · original submission · Major Revisions

We have obtained mixed review reports for the paper. It seems reasonable to offer a chance of revision to address the comments. Please provide a detailed response letter. Thanks.

Reviewer 1 ·

Basic reporting

The presentation quality of this paper is good. The background, the task, and the model are introduced clearly. However, the technical contribution of this paper is less. It seems that this paper just simply applies the VGG-19 model for solving the Zophobas Morio and Tenebrio Molitor classification problem.

Experimental design

1. It seems that the authors do not perform baselines on the dataset and provide comparisons. Specifically, what is the performance of non-transfer-learning models on the dataset? What is the superiority of the VGG-19 model compared with other transfer learning approaches on the given dataset?

2. Since the dataset is very small, the authors are suggested to use cross-validation to avoid the impact of sampling.

Validity of the findings

The novelty of this paper has not been assessed. The dataset of this paper has not been provided.

Reviewer 2 ·

Basic reporting

This is a good technical paper.
Figures are clear and have at least 300 dpi.

On the related works, the authors should move Table 2 to the Discussions section as the benchmarking between their proposed methods and other papers.

Table 2 should discuss the summary of the work, methods, strengths, and weaknesses of other papers.

Experimental design

The use of VGG-19 should be justified.
The dataset is appropriate. No data augmentation performed?

Validity of the findings

The original Table 2 could be moved here to improve proposed method validation.

·

Basic reporting

1. The English article is written quite clear and professionally.
2. The background and context are sufficient for this topic.
3. The article structure is quite reasonable
4. Relevant results to the hypothesis are quite self-contained.
5. Formal results are clear and have detailed proofs.

Experimental design

1. This article is within the aims and scope of the journal.
2. The research questions are well defined and fill the specific problems of difficulties to diffenetiate these two worms.
3. This investigation was performed to a high technical and ethical standard.
4. The methods are described with sufficient detail and information to replicate.

Validity of the findings

1. This article gives impact and novelty for worms recognition algorithm.
2. All underlying data have been provided and robust, statistically sound, and controlled.
3. Conclusions are well stated, linked to original research questions and limited to supporting results.

Additional comments

1. In general, we are not confused by distinguishing these two worms in my lab.
2. The app will help beginners to differentiate these two worms.

---

## Round 0.2 · accepted · Accept

The paper can be accepted. Congratulations.

Reviewer 2 ·

Basic reporting

I believe the authors have improved the paper based on the previous reviewer comments.

Experimental design

The authors have justified the use of VGG16. They even added other traditional classifiers and another deep learning classifier, such as Inception v3. Cross-validation has been evident as well in this revised paper.

The justification of the authors to not use data augmentation is rather acceptable.

Validity of the findings

The findings have been properly validated and benchmarked. The revised paper looks better than the original.

Additional comments

The previous issues raised have been addressed by the authors.